# Environmental Risk Assessment of Silver Nanoparticles in Aquatic Ecosystems Using Fuzzy Logic

**Rosember Ramirez** *, **Vicenç Martí** and **Rosa Mari Darbra**

Resource Recovery and Environmental Management (R2EM), Department of Chemical Engineering, Universitat Politècnica de Catalunya, 08028 Barcelona, Spain; vicens.marti@upc.edu (V.M.); rm.darbra@upc.edu (R.M.D.)
* Correspondence: rosember.ramirez@upc.edu; Tel.: +57-323-324-6890

**Abstract:** The rapid development of nanotechnology has stimulated the use of silver nanoparticles (AgNPs) in various fields that leads to their presence in different ecosystem compartments, in particular aquatic ecosystems. Several studies have shown that a variety of living organisms are affected by AgNPs. Therefore, a methodology to assess the risk of AgNPs for aquatic ecosystems was developed. The methodology is based on fuzzy logic, a proven method for dealing with variables with an associated uncertainty, as is the case with many variables related to AgNPs. After a careful literature search, a selection of relevant variables was carried out and the fuzzy model was designed. From inputs such as AgNPs' size, shape, and coating, it is possible to determine their level of toxicity which, together with their level of concentration, are sufficient to create a risk assessment. Two case studies to assess this methodology are presented, one involving continuous effluent from a wastewater treatment plant and the second involving an accidental spill. The results showed that the accidental spills have a higher risk than WWTP release, with the combination of Plates–BPEI being the most toxic one. This approach can be adapted to different situations and types of nanoparticles, making it highly useful for both stakeholders and decision makers.

**Keywords:** silver nanoparticles; risk assessment; aquatic ecosystems; fuzzy logic



## 1. Introduction

Nanoparticles (NPs) of metals, metal oxides, or metal-based compounds (such as AgNPs) exhibit remarkable biological, optical, magnetic, electronic, and catalytic properties that are typically related to their size, shape, composition, crystallinity, and particle structure. These properties have attracted a large amount of scientific and technological interest as they entail many potential applications and uses in functional materials and devices [1–3].

Silver nanoparticles, in particular, have attractive physicochemical properties, such as high electrical and thermal conductivity, and through their high biocidal activity they can suppress pathogenic microbial activity [4]. Furthermore, they can have different morphologies (spheres, rods, and cubes) [5,6] and present different coating agents or stabilizers (e.g., PVP-polyvinylpyrrolidone or citrate) [7,8]. These diverse properties make AgNPs suitable for use in a wide range of new commercial and technological applications [4,9], which has already led to an increase in both their production and their release into the environment.

In general, silver nanoparticles can be released directly and indirectly into the environment throughout their life cycle (manufacture, transport, use, and disposal) [10–12]. An example of direct release could be discharges from transport accidents and all types of spills. The indirect release could be due to discharge from wastewater treatment plants (WWTP) which receive discarded nanoparticles at the end of their life cycle. Figure 1 shows the arrival of AgNPs to different environmental compartments as a result of their release.

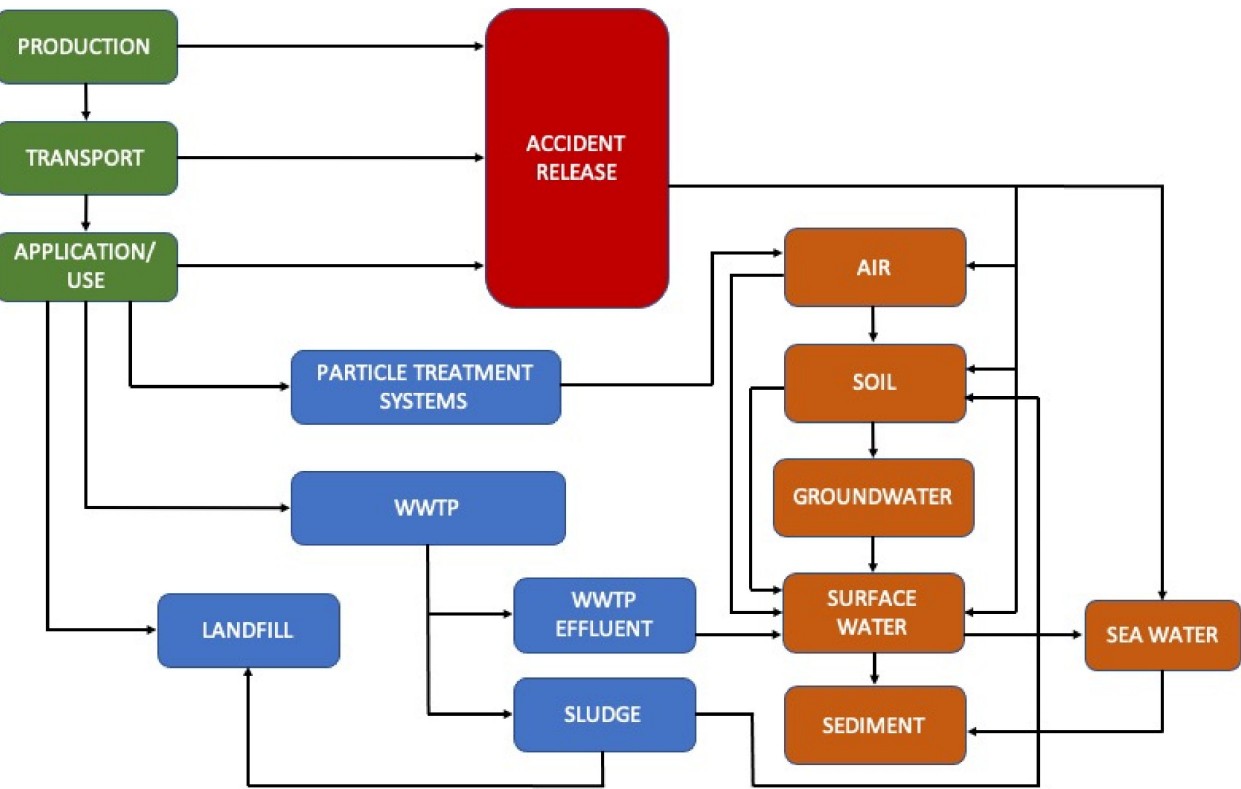

**Figure 1.** AgNPs' release pathways and associated impacts on the environment (adapted from [12]). Production/transport/application-use (green color), technical compartments (blue color), and the environmental compartments (brown color).

The surface water compartment (rivers, seas) receives the discharge of AgNPs either directly from WWTP effluents or from other environmental compartments. A study carried out in Denmark indicated that about half of the AgNPs released into the environment reach surface waters (20.5% rivers and 29.7% seas) [12]. The remainder is almost entirely deposited on land (mostly as sludge) while about 1% is released into the atmosphere. Hence, aquatic ecosystems receive an important amount of the released AgNPs.

Silver nanoparticles can also be released into the environment through accidental spills during production and transportation processes (Figure 1) [11]. However, there is very little data on such events which is why the authors decided to make this aspect one of the focal points of this paper, along with the more commonly analyzed continuous releases (i.e., WWTP) and assess the risks these events pose for aquatic ecosystems.

Table 1 reviews the concentration of AgNPs in WWTP effluents and surface waters complementing previous data from Sanchis [13]. As can be seen, the estimated AgNPs concentrations in WWTP effluents vary from 0.012 to 17,000 ng $L^{-1}$, which exceeds the measured range (0 to 1700 ng $L^{-1}$) by one order of magnitude. In surface waters, the difference is considerably larger with estimates ranging from 0–10,000 ng $L^{-1}$ while actual measurements are between 0.3–8.6 ng $L^{-1}$. Not surprisingly, measured values are lower overall in surface waters compared to WWTP effluents.

**Table 1.** Measured and estimated AgNPs in situ concentrations in WWTP effluents and surface waters for different locations.

| Data Type | Concentration (ng L$^{-1}$) | Location | Reference |
|---|---|---|---|
| WWTP Effluents | | | |
| Estimated | 32.9–11 | Europe | [14] |
| Estimated | 16.4–74.7 | EE. UU | [14] |
| Estimated | 29.8–127 | Switzerland | [14] |
| Estimated | <0.5–12.7 | United Kingdom | [15] |
| Estimated | 16.4–17000 | Global | [16] |
| Estimated | 0.012–59 | Denmark | [12] |
| Measured | 2.7–12.7 | United Kingdom | [17] |
| Measured | 0.7–11 | Germany | [18] |
| Measured | 1000–1700 | Netherlands | [19] |
| Measured | 13 | Netherlands | [20] |
| Measured | 0–7.2 * | Spain | [13] |
| Surface Waters | | | |
| Estimated | 0.588–2.16 | Europe | [14] |
| Estimated | 0.088–0.428 | EE. UU | [14] |
| Estimated | 0.555–2.63 | Switzerland | [14] |
| Estimated | 0.088–10,000 | Global | [16] |
| Estimated | 0–0.044 | Denmark | [12] |
| Measured | 2–8.6 | Isar River, Germany | [18] |
| Measured | 0.5–1.3 | Chiemsee Lake, Germany | [18] |
| Measured | 0.3–6.6 | Meuse River, Netherlands | [21] |
| Measured | 0.3–2.5 | IJssel River, Netherlands | [21] |
| Measured | 30 | Dommel River, Netherlands | [20] |
| Measured | 0.4–0.9 * | Besòs River, Spain | [13] |
| Measured | 0.4–0.7 * | Ebro River, Spain | [13] |

* In this article, the authors measured the concentration of AgNPs in L$^{-1}$ which is transformed to ng L$^{-1}$ using: $1 \times 10^8$ AgNPs L$^{-1}$ = 2 ng L$^{-1}$.

NP toxicity strongly depends on the physical and chemical properties of the core and shell [22]. In particular for AgNPs, the toxicity depends on chemical composition, size, shape, specific surface area, surface charge, and crystalline structure [23,24]. These properties play a decisive role regarding the fate of the nanoparticles in the medium and with regard to their toxicity. While the toxicity of classic contaminants typically depends on the type of contaminant itself, the toxicity of AgNPs depends on their properties, which makes the toxicity assessment much more complicated [23]. Even at the low concentrations found in aquatic ecosystems, silver nanoparticles can be toxic to various aquatic organisms [25]. Different invertebrates in the aquatic environment are affected by the biological interaction with AgNPs [26]. The toxicity is well documented and can affect fish [27,28], algae [29,30], and crustaceans [6,31]. Table 2 provides a summary of selected studies investigating the effects of AgNPs on aquatic organisms focusing on AgNPs' shape, size, and coating. For nanowires (NWs), the diameter and length (L) are specified.

**Table 2.** Impact of AgNPs properties on the toxicity of crustaceans, fish, and algae.

| Organism | Shape * | Size ** (nm) | Coating Type *** | End Point | Concentration µg/L | Conclusions | References |
|---|---|---|---|---|---|---|---|
| *Daphnia magna* | Sph | 11 | BPEI | EC50 Mortality 48 h | 0.41 | Regarding coating type, toxicity levels were as follows: BPEI > Citrate > PVP. BPEIs caused significantly higher daphnid mortality, whereas PVP-AgNPs resulted in the least toxicity. | [32] |
| | | | Cit | | 2.88 | | |
| | | | PVP | | 4.79 | | |
| *Daphnia magna* | Sph | 40 | Cit | EC50 Mortality 24 h | 8.9 | Citrate-coated AgNPs were more toxic than PVP-coated AgNPs. Additionally, the smaller their size the higher their toxicity. | [8] |
| | | 110 | | | 17.43 | | |
| | | 40 | PVP | | 24.97 | | |
| | | 110 | | | 38.35 | | |
| *Daphnia magna* | Sph | 56.6 | PVP | EC50 Mortality 48 h | 44.83 | NWs have the lowest toxicity. PL, with the smallest size, exhibited the highest degree of toxicity compared to other shapes. | [6] |
| | NWs | 41.3 L = 10,000 | | | 256.2 | | |
| | | 42.1 L = 20,000 | | | 247.1 | | |
| | PL | 30 | | | 27.92 | | |
| *Oryzias latipes* | Sph | 35 | N/A | LC50 Mortality 72 h | 1800 | Sph were estimated to be more than twice as toxic as NWs. | [5] |
| | NWs | L = 7400 | | | 4180 | | |
| *Danio rerio* | Sph | 20 | Cit | LC50 Mortality 96 h | 200 | Citrate-AgNPs were more toxic than PVP-AgNPs, and 20-nm AgNPs were more toxic than 100-nm AgNPs. | [27] |
| | | 100 | | | 400 | | |
| | | 20 | PVP | | 400 | | |
| | | 100 | | | 800 | | |
| *Danio rerio* | Sph | 10.1 | Cit and PVP | LC50 Mortality 96 h | 41.5 | PL induced higher toxicity than spheres, even at larger particle sizes. | [33] |
| | PL | 33.8 | | | 16.9 | | |
| *Pseudokirchneriella subcapitata* | Sph | 30 | Cit | EC50 Inhibition 48 h | 310 | Smaller AgNPs were the most toxic. | [29] |
| | | 15 | | | 75 | | |
| | | 30 | | EC50 Assimilation 2 h | 710 | | |
| | | 15 | | | 150 | | |

**Table 2.** *Cont.*

| Organism | Shape * | Size ** (nm) | Coating Type *** | End Point | Concentration µg/L | Conclusions | References |
|---|---|---|---|---|---|---|---|
| *Chlorococcum infusionum* | Sph | 57 | PVP | EC50 Mortality 72 h | 100 | Regarding shape, toxicity level was as follows: Plates > Wires > Sph. | [34] |
| | NWs | 42 L = 21,000 | | | 45 | | |
| | PL | 40 | | | 21 | | |
| *Pseudokirchneriella subcapitata* | Sph | 10 | Cit | EC50 Inhibition 48 h | 23.13 | Citrate-coated AgNPs were more toxic than those coated with BPEI, with the exception of the 10 nm BPEI AgNPs, which showed similar toxicity to the 10 nm Citrate AgNPs. | [35] |
| | | 30 | | | 38.28 | | |
| | | 70 | | | 118.1 | | |
| | | 10 | BPEI | | 22.92 | | |
| | | 30 | | | 67.10 | | |
| | | 70 | | | 307.4 | | |

* Shape: Sph = Sphere; NWs = NanoWires; PL = Plates; ** Size is the diameter of the Sph, NWs, or dimension of the PL. L = Length of NWs. *** Coating: PVP = Polyvinylpyrrolidone; Cit = Citrate; BPEI = Branched polyethyleneimine.

Due to the inherent uncertainty associated with some properties used to evaluate the toxicity of AgNPs, an alternative approach is required to conduct risk assessments. Fuzzy inference processes are a useful tool capable of handling these uncertainties while providing results that can be used for a more reliable systematic evaluation of associated risk factors [36–38]. A model based on fuzzy logic was developed to assess the risk of AgNPs in aquatic environments. The method was tested on two case studies: AgNPs released from a WWTP and an accidental AgNPs spill.

In order to conduct this research, a new method to evaluate the risk of AgNPs has been developed using fuzzy logic. It is a type of multi-valued logic that represents a way of addressing uncertainty and vagueness and is an alternative to classic or Aristotelian logics [37]. Whereas in classic logic one fact is true or not true, for fuzzy logic an affirmation is never totally true or false, instead of that it will be true or false with a certain degree of membership [8]. To address environmental problems, which generally involve several conflicting variables, fuzzy logic is very appropriate since it can deal with the uncertainty associated with them and provide qualitative output (e.g., a water quality index, environmental risk, etc.). Examples of applications of fuzzy logic to pollution of aquatic environmental scenarios can be found in [36,37,39,40].

The main objective of the research is to evaluate the environmental risk of silver nanoparticles in aquatic ecosystems through fuzzy logic. To achieve this objective, different tasks were conducted: modelling of the presence of AgNPs in aquatic ecosystems, study on the relevant AgNP properties that affect toxicity for aquatic organisms, development of the fuzzy logic model, and assessment of different case studies.

## 2. Materials and Methods

### 2.1. Fuzzy Logic Model

Fuzzy logic has been used successfully in environmental science to deal with uncertainties in data [36–38,41–43]. There are different procedures to implement the fuzzy principles with the Fuzzy Inference System (FIS) being the most common, capable of assigning output variables to input variables using fuzzy logic [44,45]. MATLAB (v. R2020b, The Mathworks, Inc., Natick, MA, USA) and its Fuzzy Logic Toolbox (v. R2020b) were used to carry out the analysis. The main steps in implementing a fuzzy model with FIS are:

➢ Identification of the system variables/inputs (e.g., pollutant concentration, toxicity, coating);
➢ Fuzzification by establishing fuzzy sets (e.g., high, medium, low) as well as membership functions and ranges for each variable;
➢ Use of a Fuzzy Inference Process by establishing fuzzy propositions or rules used to connect the inputs of the problem with the output;
➢ Defuzzification to obtain the final output: risk assessment.

There are two FIS or variable groups that converge on a final value: the risk (Figure 2). After a detailed literature review, the variables of AgNP shape, size, and coating were selected to determine toxicity (FIS 2 in Figure 2; Table 2 shows the relation between AgNP toxicity for different aquatic organisms and each of the above variables). Once the toxicity is established, the risk can be quantified based on the pollutant's concentration (FIS 1 in Figure 2).

The resulting model can be used for quantitative risk assessments of silver nanoparticles in aquatic ecosystems. A description for each of the selected variables is presented below (Table 2).

In general, in terms of size, the same trend is followed for each of the aquatic organisms; however, for shape and coating the trend is different. It should be noted that the observations for *Daphnia magna* were prioritized to assess the effects of shape and coating because this species is the most sensitive and presents the most coherent data.

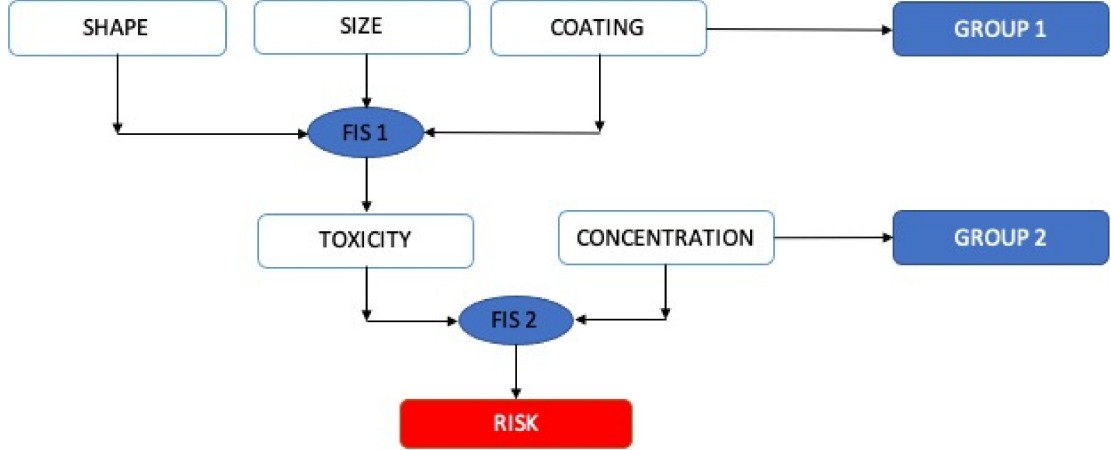

**Figure 2.** Schematic to illustrate the variable definitions and their relationships (structure of the fuzzy model) for the risk assessment of AgNPs in aquatic ecosystems.

Toxicity: This is defined as the level of damage a substance can cause in an organism [31]. Here, toxicity is an output variable and depends on the shape, size, and coating of silver nanoparticles. Toxicity has been evaluated considering the inverse of the concentration linked to endpoints. If a lower concentration produces effects on an organism, it means that this substance is very toxic.

Shape: Influences the toxicity of silver nanoparticles [5,46]. AgNPs exist as spheres, wires, and plates of different diameters. The highest toxicity is attributed to plates, followed by spheres and wires [6,33].

Size: Affects the absorption of nanoparticles by aquatic organisms [47]. Particle size is considered one of the most important factors for particle toxicity, with smaller particles being more toxic [8,29].

Coating: Is used to improve nanoparticle biocompatibility, stability, and agglomeration, which affects its toxicity [48]. Polyvinylpyrrolidone (PVP), citrate, and BPEI (Branched polyethyleneimine) are commonly used coatings. For Daphnia magna, BPEI appears as the most toxic coating type, followed by citrate and PVP [7,35] (Table 2).

Concentration: Concentrations are determined based on analytical models and studies presented in the Supplementary Material (Section S2: Transport models of AgNPs in the river). AgNP concentrations below 0.41 μg L$^{-1}$ appear to have a low impact on aquatic organisms (Table 2). Therefore, this value was used as a threshold to create the different concentration ranges for the fuzzy model (Table 2).

Risk: In general, the overall risk consists of the probability of pollutant release/spill, the exposure potential, and the level of toxicity [38]. Here, risk is calculated based on pollutant concentration and the toxicity level. Concentration is treated as a variable while toxicity is calculated based on AgNPs' shape, size, and coating.

Once the variables have been identified, the next step in the fuzzy logic methodology is defining the fuzzy sets, their ranges, and membership functions. Table 3 presents the fuzzy sets that were used in this study. As it can be seen, different typologies were selected. For example, for size, five fuzzy sets were chosen (e.g., very small, small, medium, big, and very big), each one related to a particle size range.

The maximum value was established at 450 nm since the water samples are, in general, passed through filters with a 0.45 μm pore size. The distribution between the different fuzzy sets was done following an equitable criterion. Another example is the concentration where the three fuzzy sets were established (e.g., high, medium, low). The ranges of concentrations associated to these fuzzy sets are related to the aforementioned threshold of 0.41 μg L$^{-1}$ coming from Table 3. In accordance with existing studies [36], the selected membership functions are: *Z shape*, for low values; *Pi shape*, for medium values; and *S shape*, for high values.

**Table 3.** Fuzzy sets, ranges, and types of membership function (MF) of the model variables.

| Variables | Fuzzy Set | Ranges | MF Types |
|---|---|---|---|
| Shape * | Wires | 0–5 | Z Shape |
| | Spheres | 2.5–7.5 | Pi Shape |
| | Plates | 5–10 | S Shape |
| Size ** | Very Small | 5–30 nm | Z Shape |
| | Small | 15–45 nm | Pi Shape |
| | Medium | 35–75 nm | Pi Shape |
| | Big | 65–95 nm | Pi Shape |
| | Very big | 80–450 nm | Z Shape |
| Coating * | PVP | 0–5 | Z Shape |
| | Citrate | 2.5–7.5 | Pi Shape |
| | BPEI | 5–10 | S Shape |
| Toxicity * | Low | 0–0.5 | Z Shape |
| | Medium | 0.2–0.8 | Pi Shape |
| | High | 0.5–1 | S Shape |
| Concentration | Low | 0–500 ng L$^{-1}$ | Z Shape |
| | Medium | 250–750 ng L$^{-1}$ | Pi Shape |
| | High | 500–1000 *** ng L$^{-1}$ | S Shape |
| Risk * | Very Low | 0–0.25 | Z Shape |
| | Low | 0–0.5 | Pi Shape |
| | Medium | 0.25–0.75 | Pi Shape |
| | High | 0.5–1 | Pi Shape |
| | Very High | 0.75–1 | Z Shape |

* These variables are qualitative (without specific units). ** Particle sizes were limited to <450 nm since the water samples are, in general, passed through filters with a 0.45 μm pore size. *** For values higher than 1000 ng L$^{-1}$ the function values are 1.

## 2.2. Case Studies

The Besòs river is 17.7 km in length and flows into the sea near Barcelona (northeastern Spain). Its flow rate is highly variable, typical of a Mediterranean regime (average 1.7 m$^3$ s$^{-1}$) [49]. The case study focusses on the lower 9km where the river flows through the Besòs river public park (Figure 3a) [50].

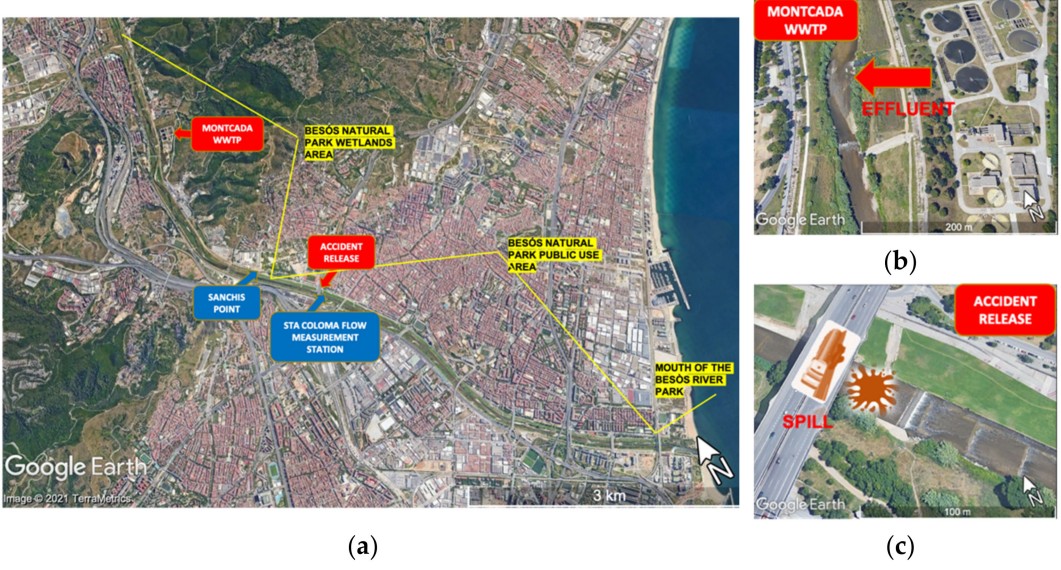

**Figure 3.** (**a**) Map of the area for the case study. (**b**) Site where effluents from the Montcada WWTP enter the river. (**c**) Site of accidental release. Images courtesy Google Earth (accessed 15 February 2022 and available in https://earth.google.com/).

This park was created in response to historical pollution and degradation problems along the Besòs river as a result of strong population growth and industrialization in adjacent areas. The park is divided into three zones: 1—Besòs natural park wetlands area, about 3 km long, mainly consisting of river meadow, islands, meanders, and 60 plots of wetland area. 2—Besòs natural park public use area, 5 km accessible to the public, mostly consisting of grassy riverbanks. 3—Mouth of the Besòs river park is of ecological significance and is restricted to public use [50].

The developed fuzzy logic model was applied to quantify the risk level posed by the presence of silver nanoparticles following a controlled discharge of effluent by a WWTP (Figure 3b) and an accidental spill (Figure 3c).

In the first scenario, the impact of effluent released from the Montcada WWTP was assessed. The concentration released by the WWTP was calculated from a measurement made by [13] (see Sanchis point in Figure 3a and Section 3 for calculation details). No attenuation mechanisms were assumed and, as a result, the pollutants became perfectly mixed both vertically and laterally with constant concentration between the discharge point and the river mouth.

The same AgNP particle size distribution as in [13] was used. In this paper, WWTP effluent data was analyzed in an area with characteristics similar to the Besòs River, and it was found that most nanoparticles were in the size range of 14–18 nm. Therefore, an average AgNPs size of 16 nm was used in the present model.

In the absence of any in situ observations, the model was run with three coating–shape combinations: citrate-coated spherical nanoparticles, BPEI-coated plate-shaped nanoparticles, and PVP-coated wire-shaped nanoparticles. This approach was followed to span different risks from the same AgNP concentration.

In the second case study, we simulated an accidental release of nanoparticles near the Santa Coloma de Gramenet measurement station (see Sta. Coloma Flow Measurement Station in Figure 3a) about 5 km from the river mouth, as a result of a road accident with a truck overturning on a bridge (Figure 3c). As a consequence of the accident, a 200 L drum of AgNP colloidal solution containing citrate-coated spheres of AgNPs with an average diameter of 10 nm at a concentration of 100 mg $L^{-1}$ is released into the river (total of 20 g of AgNPs) [51] during an approximately 17 min timespan.

For the design of both case studies, modelling used for real-time detection parameters has been taken into account for continuous release [13,52,53] and for accidental release [54–56]. These analytical models are based on advection–dispersion mechanisms [54,57,58]. Table 4 shows a summary of the parameters used in both case studies (see Section S2 in Supplementary Material for more details including justifications of the input parameters). Public data on WWTP and river flows has been used in the case studies [59–61]. As can be seen for the second case study, a replica with BPEI-coated plates was made to serve as a comparison to the first case study.

**Table 4.** Variables used for the case studies including the sections where the parameters are explained in more detail.

| | Variables | WWTP Effluents | Accidental Spill | Section |
|---|---|---|---|---|
| | | Analytical Model (see Supplementary Materials) | | |
| Source | $V_O$ (L) | N/A | 200 | S2.1 Supplementary material |
| | $t_0$ (s) | Continuous | 1000 ** | |
| | $Q_S$ (m³ s⁻¹) | 0.6–0.84 * | 0.0002 | |
| | $M$ (kg) | N/A | 0.02 | |
| | $C_S$ (ng L⁻¹) | 3.8 | $10^8$ | |
| River | $Q_R$ (m³ s⁻¹) | 2–52 * | 2.473 | S2.2 Supplementary material |
| | $h$ (m) | N/A | 0.14 | |
| | $W$ (m) | N/A | 29 | |
| | Section (m²) | N/A | 4.1 | |
| | $D_L$ (m² s⁻¹) | N/A | 402 | |
| | $v$ (m s⁻¹) | N/A | 0.61 | |
| | $x$ (m) | N/A | 5000 ** | |

**Table 4.** *Cont.*

| | Variables | WWTP Effluents | Accidental Spill | Section |
|---|---|---|---|---|
| | | Fuzzy Logic | | |
| AgNPs | Size (nm) Shape (spheres, wires, plate) Coating (PVP, citrate, BPEI) | 16 Spheres–Citrate; Plates–BPEI;Wires–PVP | 10 Spheres–Citrate;Plates– BPEI; | 2 Article |

Variables: $V_O$ = volume of the spill; $t_0$ = time during which the volume is released; $Q_S$ = outflow rate from the source; $M$ = the mass of the released AgNPs; $C_S$ = contaminant concentration at the source; $Q_R$ = river flow rate upstream from the source; $h$ = river average depth; $W$ = average width of the river; $D_L$ = longitudinal dispersion; $v$ = average flow velocity of the river; $x$ = downstream distance. * See Figure S1 in Supplementary Material for more detail. ** See Figure S2 in Supplementary Material for more detail.

### 2.3. Sensitivity Analysis

Sensitivity analysis allows for the ranking of importance of input parameters based on their relative contributions to model output uncertainty and variability [62]. Sensitivity ratio ($SR_{YX}$) is one of the common metrics of sensitivity analysis and, for quantitative parameters, measures the change in model output per unit change (($Y_2 - Y_1)/Y_0$) in an input variable ($2\Delta X/X_0$).

$$SR_{YX} = \frac{(Y_2 - Y_1) \cdot X_0}{Y_0 \cdot (X_2 - X_1)} = \frac{(Y_2 - Y_1) \cdot X_0}{Y_0 \cdot 2\Delta X} \tag{1}$$

where $Y_0$ is the output of $X_0$ (reference case). $Y_2$ and $Y_1$ correspond to the outputs of the inputs $X_0 + \Delta X$ and $X_0 - \Delta X$, respectively. For the calculation of $SR_{YX}$, only a single input $X$ at a time is modified, maintaining the rest of the parameters from the reference case constant. In the present study, the $SR_{YX}$ has been classified as Low (<0.05), Moderate (0.05–<0.2), High (0.2–<0.6), and Very High (0.6–<1), following the values used by Ferraro [63].

The model developed in the present work (Figure 2) decomposes the risk ($R$) in two main independent variables: toxicity and concentration. Toxicity is a function of size, shape, and coating variables. This approach means that toxicity sensitivity can be calculated by varying the sizes of particles for different shapes and coating scenarios, and risk sensitivity could be evaluated by performing different calculations within a range of concentration values close to the reference case.

If the range of concentrations is broad, risk sensitivity could be evaluated in several intervals of concentration from the slopes $\Delta R/\Delta C$, where $R$ is risk and $C$ the concentration. These slopes could be measured around one point ($C_0, R_o$) and can be related to $SR_{RC}$ using the following expression from reference [62]:

$$SR_{RC} = \frac{\Delta R}{\Delta C} \cdot \frac{C_o}{R_o} \tag{2}$$

Section S3 from Supplementary Materials develops sensitivity analysis of toxicity and risk for the case studies.

### 3. Results and Discussion

#### 3.1. Montcada WWTP Effluents Case Study

Based on the analysis conducted in this paper, the plate-shaped BPEI-coated AgNPs yielded the greatest risk (medium risk) compared to the other two combinations. Table 5 shows the inputs for each variable and the outputs obtained once the fuzzy model was applied. The first step is to achieve the toxicity value from the inputs size, shape, and coating. From here, the second step is to combine this toxicity with the concentration to obtain the risk value.

**Table 5.** Input variables and output (risk) for the different particle types in the WWTP controlled effluent release case study.

| | WWTP Particle Types | | |
| --- | --- | --- | --- |
| **Variable** | **Spheres Citrate** | **Plates BPEI** | **Wires PVP** |
| Size (nm) | 16 | 16 | 16 |
| Shape * | 5 | 10 | 0 |
| Coating * | 5 | 10 | 0 |
| Toxicity * | 0.515 | 0.855 | 0.145 |
| Concentration in the river (ng L$^{-1}$) | 0–3.7 ** | 0–3.7 ** | 0–3.7 ** |
| Risk (fuzzy value) * | 0.25 | 0.5 | 0.07 |
| Risk (fuzzy qualitative value) | 100% Low | 100% Medium | 83% Very low–17% Low |

* These variables do not have units since they are qualitative. ** Annual range of concentration, see Figure S1 in supplementary material.

The spherical citrate-coated AgNPs resulted in a low risk, although they have a medium toxicity level. The lowest risk (very low to low) is posed by wire-shaped PVP-coated AgNPs which already have a much lower inherent toxicity compared to the other two particle types.

Section S3 of Supplementary Materials is focused on sensitivity analysis. For this case study, the toxicity and the risk of the developed model are not sensitive to size (see Tables S1 and S2 in Supplementary Materials) but are very sensitive to the type of AgNPs, that is considered unknown. The concentration in the river varies yearly in a very narrow range, which does not affect the risk assessment results after testing it with the appropriate tool. Instead, it is the toxicity linked to shape and coating that determines the level of risk. The concentrations used in the simulation are too low to result in any significant change in risk level.

*3.2. Accidental Release Case Study*

In this second case study, two particle types were considered: citrate-coated spheres (based on specifications from a local manufacturer) and BPEI-coated plates (to serve as a comparison to the first case study). In contrast to the first case study, here the concentration is variable in both time and space. As in the previous case study, Table 6 shows the inputs for each variable and the outputs obtained with the fuzzy model. The toxicity value comes from the size, shape, and coating inputs, whereas the risk value comes from the combination of the toxicity and concentration values. Spherical citrate-coated particles resulted in a high risk level, while BPEI-coated plates reached a high to very high risk level.

For spherical citrate-coated particles, the time course exhibits an increase that reaches a maximum (100% high risk) after about 5000 s (around 1 h 25 min) before decreasing again at 9500 s (2 h 40 min approx.) (Figure 4a). In the case of BPEI-coated plates, the maximum level (approx. 80% very high risk and 20% high risk) is achieved at the same time with an earlier decrease (9000 s) (Figure 4b).

The 100% high risk level is achieved at a concentration of 695 ng L$^{-1}$ for the BPEI-coated plates, while the corresponding concentration for citrate-coated spheres is slightly higher (725 ng L$^{-1}$). However, for the BPEI-coated plates, the risk at maximum concentration increases to very high risk level, while in the case of citrate the risk is maintained to high risk. This proves that BPEI-coated plates' particles are more dangerous for the aquatic environment than the citrate-coated spheres, as presented in Table 6.

With reference to the sensitivity of the model, in the accidental cases, as in the case of WWTPs, the toxicity and the risk of the developed model were insensitive to size (see Table S3 and Figure S3 in Supplementary Materials). The toxicity and risk have low sensitivity to the type of AgNPs because it is assumed to be known, and the sensitivity

of risk depends on specific concentrations ranging from insensitive to Very High SR (see Table S4 in Supplementary Materials).

**Table 6.** Input variables and output (risk) for the different particle types in the accidental spill case study.

| | Accidental Spill Particle Types | |
|---|---|---|
| **Variables** | **Spheres Citrate** | **Plates BPEI** |
| Size (nm) | 10 | 10 |
| Shape * | 5 | 10 |
| Coating * | 5 | 10 |
| Toxicity * | 0.515 | 0.855 |
| Concentration in the river (ng L$^{-1}$) | 0–1000 ** | 0–1000 ** |
| Risk (fuzzy value) * | 0.76 | 0.93 |
| Maximum Risk (fuzzy qualitative value) | 100% High | 81.5% Very High– 18.5% High |

* These variables do not have units since they are qualitative. ** See Figure S2 in Supplementary Material for more detail.

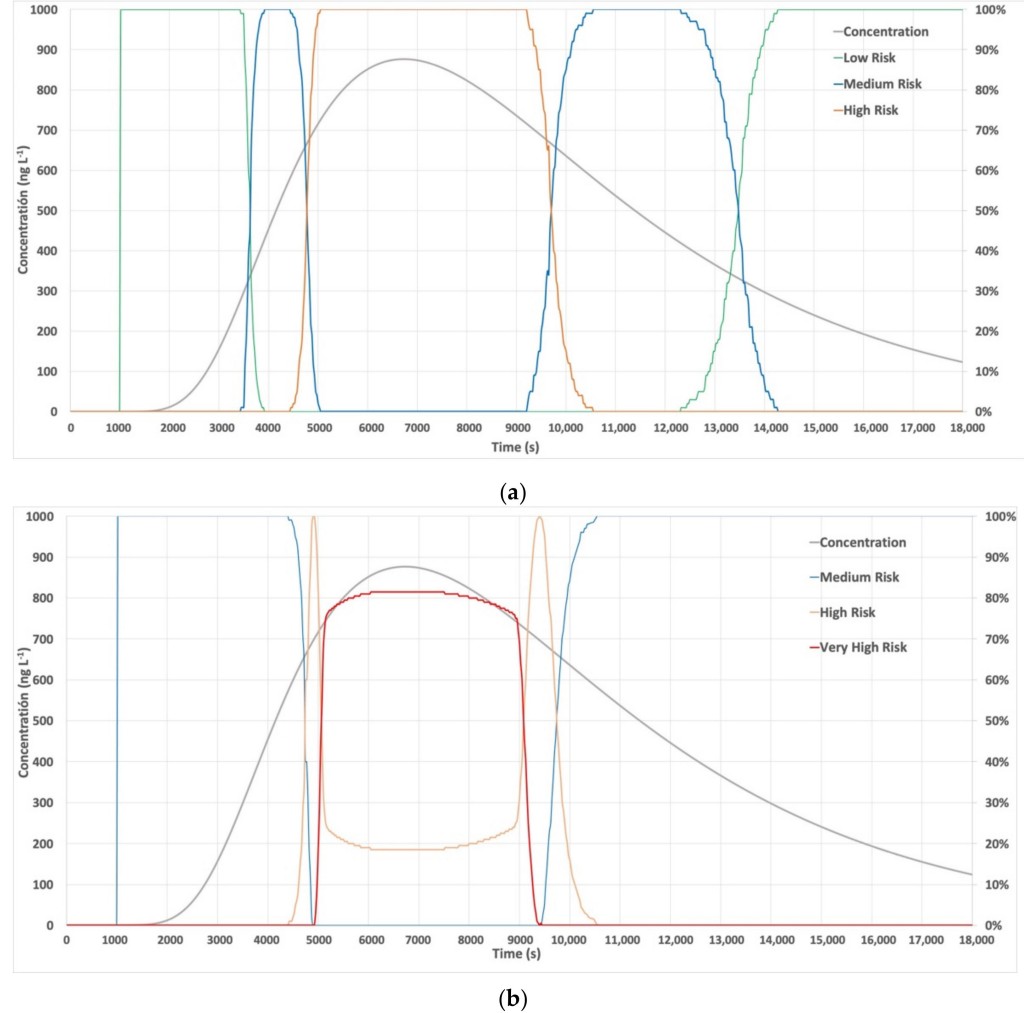

**Figure 4.** Time course of the risk levels for (**a**) spherical citrate-coated particles and (**b**) BPEI-coated plates for the accidental spill scenario. Time zero corresponds to the time of the accidental spill.

Comparing both case studies, AgNPs exhibit different risk behavior. In the controlled release from the WWTP, risk varied from very low to medium, whereas in the accidental spill it varied from low to very high. While citrate-coated spheres yielded a 100% low risk assessment in the WWTP scenario, in the accidental spill a high risk level was obtained. BPEI-coated plates yielded a 100% medium risk assessment in the WWTP scenario while reaching a maximum of 81.5% very high risk in the accidental spill scenario. Hence, although particles had the same toxicity values in both scenarios, the resulting risk level was different due to differences in concentration.

The toxicity values carry a lot of uncertainty as they depend on other characteristics such as size, shape, and coating which may be unknown. Therefore, using fuzzy logic to develop the presented method has been of great assistance. This is in agreement with the study carried out by [38].

## 4. Conclusions

This paper assesses the environmental risk of silver nanoparticles in aquatic ecosystems using fuzzy logic. Different levels of toxicity and risk have been obtained, ranging from low to very high according to the scenario, type of particles (size, shape and coating), and concentration.

In the first case study, the release from a WWTP, Plates–BPEI AgNPs were the most toxic, followed by Spheres–Citrate and Cables–PVP.

In the second case study, the accidental spill, the concentration varies as a function of the time from the accident show a gaussian shape. As a consequence, different risk levels were obtained for each type of particle. Maximum risk levels were reached once the concentration was varied. In the same way as in the first case study, the Plates–BPEI generated a higher risk compared to the Spheres–Citrate. Although in this case the risk levels are much higher than for the WWTP.

In both case studies, the greater the toxicity the greater the risk. In addition, BPEI-coated plates posed the greatest risk and PVP-coated wires the lowest.

In general terms, the toxicity and the risk of the developed model are not sensitive to size. In the case of the WWTP scenario, the toxicity and the risk are very sensitive to the type of AgNPs, but risk is insensitive to concentration variations. In the accidental cases, the toxicity and the risk have low sensitivity to the type of AgNPs and the sensitivity of risk depends on specific concentrations, ranging from insensitive to Very High SR.

Using fuzzy logic has allowed for the combination of several uncertain factors related to the risk of AgNPs and a final risk assessment to be provided. The tool can be adapted to other types of NPs and environments, which makes it very appropriate for environmental decision makers.

**Supplementary Materials:** The following supporting information can be downloaded at: https://www.mdpi.com/article/10.3390/w14121885/s1, Figure S1: River flow and AgNPs concentration in Besòs river between the source and 5 km downstream from the discharge point. Figure S2: AgNPs concentration over time, 5 km downstream from the accidental spill. Table S1: Sensitivity analysis for Toxicity (WWTP). Table S2: Sensitivity analysis for Risk (WWTP) for the range of concentration 0.25–4.75 ng $L^{-1}$. Table S3: Sensitivity analysis for Toxicity (Accidental release). Figure S3: Accidental Risk as a function of concentration. (a) Spheres–Citrate; (b) Plates–BPEI. Table S4: Summary of Sensitivity Analysis.

**Author Contributions:** Conceptualization, R.M.D. and V.M.; methodology, R.M.D. and V.M.; investigation, R.R.; resources, R.M.D.; writing—original draft preparation, R.R., R.M.D., and V.M.; writing—review and editing, R.M.D. and V.M. All authors have read and agreed to the published version of the manuscript.

**Funding:** This research was funded by the Spanish Ministry of Science, Innovation and Universities and Agencia Estatal de Investigación/European Regional Development Plan (grant CGL2017-87216-C4-3-R). Authors want to thank Fundación Carolina and Universidad Tecnológica del Chocó for the PhD research grant given to Mr. Ramírez.

**Institutional Review Board Statement:** Not applicable.

**Informed Consent Statement:** Not applicable.

**Data Availability Statement:** The data presented in this study is available on request from the corresponding author.

**Acknowledgments:** Authors want to thank Loubna Zerrouki for the state of art about AgNPs.

**Conflicts of Interest:** The authors declare no conflict of interest.

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
