# Peer review of "Environmental Risk Assessment of Silver Nanoparticles in Aquatic Ecosystems Using Fuzzy Logic"

_water, doi:10.3390/w14121885_

Round 1

Reviewer 1 Report

The text is interesting, but it has some shortcomings. Methodology and results need to be better described. For exmaple, there is no information on how a given case is assigned to a given fuzzy set. Tables 2,4,5 need to be better described in the article. Some units are missing form the tables.

Reviewer 2 Report

The authors assessed the environmental risks of silver nanoparticles in aquatic ecosystems using fuzzy logic. The model is very novel and all the datasets used are clearly illustrated in the supplementary data. The approach used has the potential to be applied to different situations and types of nanoparticles. This manuscript can be accepted after minor correction.

Minor comments

1. Uncertainty analysis should be added for the predicted results;

2. Fig. 1 “WWTP EFLUENT” to WWTP EFFLUENT

Reviewer 3 Report

The study by Ramirez et al., entitled „ Environmental risk assessment of silver nanoparticles in aquatic ecosystems using fuzzy logic“ is an original research article dealing with a development of a fuzzy logic model and its testing in risk assesment of silver nanoparticles in aquatic ecosystems, using two model case studies.

The study has the potential for improvements.

Major concerns related to present form of the manuscript:

  1. The manuscript relies to a great extent to information provided in supplementary information, so many parts of the manuscript are difficult to understand without reading the Supplementary information in parallel. The manuscript should be entirely understandable by itself, and supplementary information should contain only some additional data.
  2. The aim of the study is not clearly presented and should be added into the manuscript.
  3. The results are not well discussed. The Results and discussion section contains only description of the results, without a serious discussion in context of the present knowledge and other available studies, so the significance and the true meaning of the results and suggested fuzzy logic model remains unclear.
  4. Conclusions contain long introduction, and the bullets represent rather general remarks. After the aim of the study is clearly defined and results are discussed, conclusions can be rewritten to represent true knew knowledge that resulted from this study.

ADDITIONAL COMMENTS:

INTRODUCTION

Lines 26 and 31 – please consider replacing the term „interesting“  with a term that would more precisely describe the properties of NPs

Line 36 – please change „interesting for use” into “suitable for use”

The third paragraph (lines 39-44) needs a revision. Namely, in most cases, nanoparticles of metals enter the environment contained in the related products,  and are being released from these products during their life cycle through product use and disposal. Since nanoparticles are in general intentionally used and produced, terms of intentional and unintentional release in the environment in a classical way of meaning should be avoided. For revising this paragraph, I suggest authors to consider the review by Gottschalk and Nowack (2011) J. Environ. Monit. 13, 1145, and provide more precise information on the release of engineered nanomaterials to the environment.

A brief description and the main principle of a fuzzy logic should be added to the last paragraph of the Introduction.

MATERIAL AND METHODS

Figure 2 – The authors could consider putting the entire scheme into the opposite direction by changing the direction of arrows: from shape, size and coating (which would then be Group 1) to FIS2 (which should then be changed into FIS1), then to toxicity (which would then be Group 2), and from toxicity and concentration to FIS1  (which should then be changed into FIS2), to finalize with the risk.  In this way, the scheme would really represent convergence to risk as a final value (as stated in line 103), instead of divergence presented by the present version, and better describe the structure of the fuzzy model.

Lines 122-123 – the term “concentration of the evaluating organism” should be revised, to correspond to the provided definition of toxicity.

Table 3 – abbreviations for variables should be defined in the table legend.

RESULTS AND DISCUSSION

Table 4 and Table 5 – from the manuscript, it is not clear how exactly the risk (both fuzzy values and fuzzy qualitative values) was calculated

Reviewer 4 Report

The authors have presented an interesting study mentioning a unique risk assessment model i.e.  fuzzy logic model; a methodology to assess the risk of silver nanoparticles (AgNPs) for aquatic organisms. The research paper is generally well written and structured. AgNPs exhibit interesting biological, optical, magnetic, electronic, and catalytic properties that are typically related to their size, shape, composition, crystallinity, and particle structure. Also, the toxicity values carry a lot of uncertainty as they depend on other characteristics which may be unknown. Therefore, using fuzzy logic to develop the presented method has been of great assistance. It allowed to combine several uncertain factors related to the risk of AgNPs and thus provided a final risk assessment. The tool can be adapted to other types of NPs and environment, which makes it very appropriate for environmental decision makers. There are some concerns that need some attention of the researchers:

(i) The authors are recommended to mention the results and some data findings in the last few lines of the abstract.

(ii) Author should briefly quote some toxic effects of AgNPs on aquatic ecosystem in the introduction part . Refer to the paper (Thimmegowda, R., 2020, Effect of Silver Nanoparticles on Aquatic Organisms.)

(iii) Authors are recommended to mention a little information on how the nanoparticles' toxicity is dependent on their physical and chemical properties. Refer to the paper: (Sukhanova, A. et al. (2018). Dependence of nanoparticle toxicity on their physical and chemical properties. Nanoscale Research Letters, 13(1), 1-21).

Reviewer 5 Report

This manuscript entitled "Environmental risk assessment of silver nanoparticles in aquatic ecosystems using fuzzy logic" by Remirez et al. showed a model to assess the accidental risk of AgNPs for aquatic ecosystems . This manuscript can be improved after revsion.

  1. Author must include more details on varification of their model in result and discussion section.
  2. More details needed on Daphnia magna that is used as sensitive species to presents the most consistent data.
  3. Author must co-related with real time detections using live analysis to support their models.
  4. Author must include more information in the introiduction part by citing all realted prior report on these kind of models for assesments on pollution in water 

Round 2

Reviewer 3 Report

The authors significantly improved the quality of the manuscript after revision.